# Silver Doped Magnesium Ferrite Nanoparticles: Physico-Chemical Characterization and Antibacterial Activity

**DOI:** 10.3390/ma14112859

**Published:** 2021-05-26

**Authors:** Erika Fantozzi, Erlinda Rama, Cinzia Calvio, Benedetta Albini, Pietro Galinetto, Marcella Bini

**Affiliations:** 1Department of Chemistry, University of Pavia, Viale Taramelli 16, 27100 Pavia, Italy; erika.fantozzi01@universitadipavia.it; 2Department of Biology and Biotechnology “L. Spallanzani”, University of Pavia, via Ferrata 9, 27100 Pavia, Italy; erlinda.rama01@universitadipavia.it (E.R.); cinzia.calvio@unipv.it (C.C.); 3Department of Physics and CNISM, University of Pavia, via Bassi 6, 27100 Pavia, Italy; benedetta.albini01@universitadipavia.it (B.A.); pietro.galinetto@unipv.it (P.G.)

**Keywords:** MgFe_2_O_4_, X-ray powder diffraction, micro-Raman, Ag doping, antibacterial activity

## Abstract

Spinel phases, with unique and outstanding physical properties, are attracting a great deal of interest in many fields. In particular, MgFe_2_O_4_, a partially inverted spinel phase, could find applications in medicine thanks to the remarkable antibacterial properties attributed to the generation of reactive oxygen species. In this paper, undoped and Ag-doped MgFe_2-x_Ag_x_O_4_ (x = 0.1 and 0.3) nanoparticles were prepared using microwave-assisted combustion and sol–gel methods. X-ray powder diffraction, with Rietveld structural refinements combined with micro-Raman spectroscopy, allowed to determine sample purity and the inversion degree of the spinel, passing from about 0.4 to 0.7 when Ag was introduced as dopant. The results are discussed in view of the antibacterial activity towards *Escherichia coli* and *Staphylococcus aureus*, representative strains of Gram-negative and Gram-positive bacteria. The sol–gel particles were more efficient towards the chosen bacteria, possibly thanks to the nanometric sizes of metallic silver, which were well distributed in the powders and in the spinel phase, with respect to microwave ones, that, however, acquired antibacterial activity after thermal treatment, probably due to the nucleation of hematite, itself displaying well-known antibacterial properties and which could synergistically act with silver and spinel.

## 1. Introduction

The great technological progresses in nanoscience extend to different fields, such as biology, medicine, chemistry, pharmacy, agriculture, food industry, and materials science among others. In recent years, biomedical applications of nanostructured biomaterials have increased remarkably [1,2,3,4]. In this regard, a very fascinating class of materials are undoubtedly the spinel ferrites, with main applications in contrast enhancement of magnetic resonance imaging (MRI), bio-magnetic separation, treatment of cancer by hyperthermia, and drug delivery and release [5,6,7]. Their advantages are numerous: improvement of pharmacokinetic and pharmacodynamics profiles of drugs, increased drug stability and solubility in aqueous phase, enhancement of accumulation in specific tissues through passive and active magnetic targeting and reduction of the drug concentration in non-targeted normal tissues and toxic side effects [7].

Spinels are also valuable for the preparation of modern sensors and biosensors, which are applicable in both industrial and biomedical areas [8,9]. In addition, in certain formulations, metals, metal oxides and carbon-based nanoparticles (NPs) have shown powerful antibacterial activities [10,11]. These effects were found to depend on a series of intrinsic and extrinsic factors. The intrinsic factors include concentration, size (the smaller the size, the greater the bactericidal activity, especially for sizes < 30 nm), shape (triangular and sharper NPs have greater activity than spherical NPs), and chemical composition. Extrinsic factors comprise environmental factors (e.g., aerobic vs. anaerobic milieu, pH), type of bacteria (depending on the structure of the cell wall), bacterial cell growth (rapidly growing bacteria are more sensitive), metabolic and cell cycle phase (e.g., planktonic vs. sessile bacteria), and the presence of an established biofilm, acting as a barrier, reducing the exposure of inner encased cells [12]. Silver NPs (AgNPs) appear to (i) interact with the bacterial cell wall disturbing its permeability; (ii) inactivate essential proteins such as thiol-containing enzymes; (iii) cause DNA condensation; and (iv) lead to reactive oxygen species (ROS) generation, with mechanisms like those described for Ag^+^ ions [13,14,15]. Furthermore, biocide Ag^+^ ions are released from AgNPs [16].

NPs can be utilized in different ways. They can be applied as a biomaterial coating, but they can also be used to dope bulk biomaterials. The surface of NPs can be modified to further improve bactericidal properties or lower toxicity, for instance using a chitosan coating [17] or by surface derivatization with alkylated polyethylenimines [18]. Nanotechnologies can be finally applied to enhance the antimicrobial action of photocatalytic surfaces or of NO-releasing surfaces, taking full advantage of the large surface/volume ratio of nanostructured materials. Concerns exist regarding the still-incomplete knowledge base of the toxicology of nanomaterials. Experimental models to investigate the toxicology of nanoparticles have not been standardized yet and lack reproducibility, often leading to inconsistent results. Induction of apoptosis and genotoxic effects and potential translocation of NPs to distant tissues/organs, with the risk of systemic effects, have also been described for materials in the form of nanoparticles.

In 2017, the World Health Organization (WHO) published a list of antibiotic-resistant “priority pathogens”, which classified 12 families of bacteria that represent the greatest risk to human health. Various Enterobacteriaceae, such as *E. coli*, were cataloged among the most critical group, i.e., the multidrug resistant bacteria (MDR). *S. aureus* is, instead, classified as bacterium with a high priority [19]. In fact, it belongs, together with *Enterococcus faecium*, *Klebsiella pneumoniae*, *Acinetobacter baumannii*, *Pseudomonas aeruginosa*, and *Enterobacter* spp., to the ESKAPE bacteria, a group of pathogens that can evade antibacterial drugs [20]. Antimicrobial resistance represents an important global health crisis, which has prompted research for alternative antibacterials, such as antibiotic adjuvants or metal-based antibacterial agents [21].

Among the different ferrites, MgFe_2_O_4_ has peculiar magnetic and physical properties that lead to its wide application in medicine. The antibacterial activity of magnesium-based nanoparticles is attributed to the generation of ROS [22,23,24,25,26].

In this work, the study of undoped and Ag-doped MgFe_2-x_Ag_x_O_4_ (x = 0.1 and 0.3) NPs synthesized using microwave-assisted combustion and sol–gel routes is reported. In addition, the doped particles were thermally treated in order to evaluate the effect of sinterization on bacterial growth inhibition. A thorough characterization of the structural, morphological, compositional, and vibrational properties has been performed by combining X-ray powder diffraction (XRPD) with Rietveld structural refinement, SEM/EDS analyses and micro-Raman spectroscopy. Sample purity, lattice parameters, crystallite sizes and the inversion degree of the spinel phase were determined. The measured physico-chemical properties of magnesium ferrites have been discussed in view of their antibacterial activity.

## 2. Materials and Methods

### 2.1. Synthesis

The undoped and doped MgFe_2-x_Ag_x_O_4_ (x = 0, 0.1 and 0.3) samples were synthesized using two different synthetic routes, microwave-assisted combustion (MW) and sol–gel (SG).

MW [27]: Mg(NO_3_)_2_·6H_2_O and Fe(NO_3_)_3_·9H_2_O, with or without the addition of Ag(NO_3_), in a stoichiometric ratio, were mixed with an amount of citric acid, used as fuel, calculated from the propellant chemistry theory. The mixtures were placed in a Milestone 1200 Pyro microwave oven for 120 min at 800 W (the temperature in the oven was about 300 °C); after cooling to room temperature, the powders were ground in a mortar. These samples were named MW, MW-1 and MW-3, according to amount of silver addition. The samples were further treated in oven at 600 °C for 4 h. After thermal treatment, the above samples were, respectively, named MW-T, MW-1T and MW-3T.

SG: Mg(NO_3_)_2_·6H_2_O and Fe(NO_3_)_3_·9H_2_O, with or without the addition of Ag(NO_3_) in a stoichiometric ratio, were mixed with citric acid (a ratio of about 2:1 citric acid:total mol amount of cations) and dispersed in water; the mixture was stirred at about 80 °C until a gel formed. Then, the solvent was completely evaporated, and the obtained powders were treated in oven at 500 °C for 4 h. Samples were named SG, SG-1 and SG-3, accordingly to their silver content. Samples were further treated in oven at 700 °C for 4 h and named SG-T, SG-1T and SG-3T.

### 2.2. Characterization Techniques

X-ray powder diffraction (XRPD) measurements were performed by using a Bruker (Karlsruhe, Germany) D5005 diffractometer with CuK α radiation (40 kV, 40 mA), a graphite monochromator and a scintillation detector. The patterns were collected in air with a step size of 0.03° and counting time of 4 s/step in the angular range 16–100°, using a silicon sample holder with low background.

Rietveld structural and profile refinement was carried out by means of TOPAS software (Version 3.0, Bruker, Karlsruhe, Germany) [28] on the basis of the known crystal structure model of the cubic spinel. During the refinement, the background coefficients, scale factor, zero error, lattice parameters, crystallite sizes, isotropic thermal factors and atomic positions were allowed to vary, as well as the occupancies, to verify the possible inversion degree of the spinel. Proper constraints were used to limit the dopant amount to its stoichiometric value and to allow the inversion on the tetrahedral and octahedral cationic sites. The weight percentages of secondary phases, when present, were also determined.

A Zeiss EVO MA10 (Carl Zeiss, Oberkochen, Germany) scanning electron microscope (SEM) coupled with an EDS detector (X-max 50 mm^2^, Oxford Instruments, Wiesbaden, Germany) was used for morphological studies and elemental microanalyses of the samples. SEM measurements were performed on gold sputtered samples.

Micro-Raman measurements were carried out at room temperature using a Labram Dilor spectrometer (Horiba, Kyo Metropolis, Japan) equipped with an Olympus microscope HS BX40. The 632.8 nm light from a He-Ne laser was employed as excitation radiation. The samples, mounted on a motorized *xy* stage, were tested with a 100× objective and with a laser spot of ~1 μm in diameter. The spectral resolution was about 1 cm^−1^. Neutral filters with different optical density were used to irradiate the samples at different light intensities leading to power density values from 5 × 10^3^ W/cm^2^ to 5 × 10^5^ W/cm^2^. A cooled CCD camera was used as a detector and the typical integration times were about 2 min. The sample phase homogeneity was verified by mapping the Raman spectra from different regions of each sample. The parameters of the Raman spectra were extracted using best fitting procedures based on Lorentzian functions. In this way the frequency, full width at half maximum, intensity and integrated intensity of the peaks were determined.

### 2.3. Antibacterial Activity

Nanoparticles were suspended in DI-H_2_O at a concentration of 100 mg/mL and sterilized by low-pressure autoclaving at 10 psi for 20 min. Two different pathogenic bacterial strains, *E. coli* ATCC 25922 [29], and *S. aureus* ATCC 25923 [30] were cultured on Luria-Bertani (LB) (10 g/L tryptone, 10 g/L NaCl, 5 g/L yeast extract, pH 7) and incubated at 37 °C for 16 h with orbital shaking. Cell density was determined with a UV-Vis spectrophotometer set at 600 nm wavelength (OD_600_). Cultures were diluted to 0.125 OD_600_ in LB and 2 mL of each culture was poured on LB-1.5% agar plates (90 mm), which were gently rotated to uniformly distribute the liquid on the surface; excess liquid was discarded, and plates were allowed to dry in a laminar flow hood. Circular wells were created on each dried plate with a sterile syringe cut at the tip, using the plunger of the syringe to suck up the medium lump. In each well, 20 μL of sterile nanoparticle solution (100 mg/mL in H_2_O) were loaded and plates was incubated in the upward position (with the lid on top) at 37 °C for 16 h in a static incubator. Each experiment was performed in biological triplicates. Images of plates were acquired through a gel documentation system using white light illumination. The freehand selection tool of ImageJ software (version 1.52c, Wayne Rasband, National Institutes of Health, Bethesda, MD, USA) [31] was used by two independent researchers to measure the area of no-growth halos. Inter-operator reliability, calculated with the Manigold Reliability Calculator [32], was above 0.98. Statistical analysis was applied to assess significance of the growth inhibition halo given by each Ag-doped material by comparison with that of the corresponding undoped nanoparticles (Student’s *t* test (two tailed; *: *p* < 0.05; **: *p* < 0.01; ns: non-significant)).

## 3. Results

### 3.1. XRPD and Rietveld Refinement

In Figure 1A,B, the diffraction patterns of MW and SG MgFe_2_O_4_ samples are shown. 

For the MW samples (Figure 1A), the patterns of doped materials were clearly constituted by two crystalline phases, the cubic MgFe_2_O_4_ spinel and the metallic Ag [22]. The reflections intensities of this last phase increased by increasing the dopant amount, as expected. A different peak broadening for the two phases was detected: the spinel had broader peaks suggesting nanometric dimensions, while the silver had narrower peaks. The thermal treatment of doped samples did not substantially modify the patterns; a moderate sinterization effect could be determined from the narrowing of the spinel phase peaks. The main peak of hematite at about 33° was evident; it was very narrow and weak, suggesting large crystallite sizes and a small amount of the phase. This indicated that the temperature increase could favor phase nucleation, probably starting from hematite nuclei already present in the sample due to a possible very limited inhomogeneity. The Ag peaks were practically unchanged after thermal treatment.

The SG XRPD patterns are reported in Figure 1B. The main difference with respect to the MW samples was that SG-1 clearly only showed the spinel phase reflections, suggesting the incorporation of Ag ions into the spinel lattice, while the SG-3 pattern was similar to MW-3, even if Ag and spinel peaks were broadened, suggesting that part of Ag could be substituted onto the spinel lattice also in this case. The SG synthesis appeared to favor the substitution of silver into the spinel cubic structure, with a possible solubility limit, without alterations. The same was not true for MW synthesis where silver, independently from its amount, was present as a separate phase. For SG samples, the Ag doping reduced the crystallinity of the starting material: the spinel peaks were broadened and with lower intensities, consistently with the presence of nanostructures. As expected, the thermal treatment produced a more prominent narrowing of reflections with respect to MW series. For SG-1T, the hematite main peak could be appreciated, as well as the peaks of silver—a phase separation can probably be assumed.

Rietveld structural and profile refinements were performed on all the patterns to determine the main structural parameters, on the basis of the known crystallographic models of magnesium spinel and metallic silver phases. It must be recalled that MgFe_2_O_4_ is an inverse spinel, with the inversion degree (I) defined as the amount of the trivalent cations on tetrahedral sites [22,25,26]. During refinements, the occupancy factors were also varied to verify the influence of nanoscale and thermal treatment on the cation distribution. In Figure 2, as an example, the results of the Rietveld refinements on MW-1 and SG-1 patterns are reported, while patterns for other samples are shown in the Appendix A, Appendix A. The difference curves (at the bottom) between the experimental and calculated patterns were almost flat, suggesting good refinements, as well as the discrepancy factor values, in particular GoFs, near to 1 (Table 1 and Table 2). The lattice parameters, crystallite sizes of spinel and silver, inversion degree and secondary phase’s amount are also reported in Table 1 and Table 2. The lattice parameters slightly increased with Ag doping and with thermal treatment [25], but they were practically the same for the MW and SG series of samples. On the other hand, silver does not substitute onto spinel cationic sites (except for SG-1 and, to some extent, for SG-3, for which a slight increase could in fact be appreciated), thus, the main differences were due to oxygen vacancy formation, to balance the lack of positive charges, producing lattice relaxation. The crystallite sizes of the spinel main phase were in the nanometric range for all samples from both the syntheses. For the MW ones, the thermal treatment influenced the crystallite size values of spinel only slightly, while a marked increase, from 8 nm to about 50 nm, was observed for the SG ones [23]. This effect could be related to the microstructure of the samples, which was more compact for MW with respect to that of SG, which was fluffier. The crystallite sizes of metallic silver, present as separated phase, were significantly different between the two series of samples. In the MW ones, values between 22 and 36 nm were obtained, slightly higher than those of the spinel phase, while for SG samples the values were about 10 nm, independent of the thermal treatment (Table 1 and Table 2). This evidence was clearly related to the different synthesis route; the citric acid used for the preparation of SG acted as a capping agent, limiting the grain increase. In addition, the Ag distribution in the two series of samples was also different; in the MW, Ag formed aggregated particles well evident between the ferrite grains (see SEM in the following paragraph), limiting the growth of the main phase grains with the temperature increase. In the SG ones, silver was more distributed, in a large part substituted onto the spinel lattice, with no evidence of aggregates, thus facilitating the sinterization of the main phase during the thermal treatment. These evidences will be useful to explain the antibacterial results.

It is interesting to observe the trend of the I parameter; in both the syntheses, the Ag doping favors the inversion degree passing from about 0.4 (for pure samples) to 0.7–0.8 (for Ag doped). No particular trend of I is, however, evident with the increase in the dopant amount. Often, in the literature, spinel inversion has been related to the reduction of particle sizes [5,6]; however, this is not the case, as almost all the synthesized materials were nanometric. Therefore, we can suppose that the presence of Ag as segregated phase caused oxygen vacancies in the spinel structure to balance the vacancies on iron sites introduced by the Ag segregation. In these conditions, i.e., with a more disordered crystal structure, the exchange between magnesium and iron ions on cationic sites could be favored, as also happens when Ag is present in the cubic lattice.

### 3.2. Morphological and Compositional Analysis

SEM images of several samples, taken with the same magnification, are reported in Figure 3; images of the remaining samples are shown in Appendix A. The MW samples appeared to be constituted by extended blocks with sharp edges. In the internal part, small, rounded particles could be recognized, suggesting that the extended aggregates are composed of nanometric particles. SG samples presented similarities with the corresponding MW ones; they appeared compacted, but with smooth surfaces. The particles, upon thermal treatment, appeared fused together, corroborating the hypothesis, based on XRPD data, of a temperature-dependent increase in size values of the spinel crystallites (Table 1 and Table 2).

EDS microanalysis allowed to verify the chemical compositions, that were in good agreement with the stoichiometric ones. The Fe/Mg ratio was two for all samples analyzed and the Ag amount was close to the substituted values (Appendix A). In Appendix A it can be seen that, for the MW-1 sample, Ag was present in the form of large, brilliant aggregates, distributed between spinel particles, as also demonstrated by the punctual EDS analysis. The same was not true for SG-1, where Ag ions appeared more distributed into the sample. These observations are in agreement with the XRPD data, particularly as far as the trend of crystallite sizes with thermal treatment is concerned.

### 3.3. Raman Spectroscopy

The cubic MgFe_2_O_4_ (s.g. Fd3m) has a full unit cell with 56 atoms (Z = 8), but the Bravais cell only contains 14 atoms (Z = 2). Group theory analysis at the center of first Brillouin zone predicts 3 acoustic modes and 39 optical modes with the following symmetries: A_1g_ + E_g_ + F_1g_ + 3F_2g_ + 2A_2u_ + 2E_u_ + 4F_1u_ + 2F_2u_ [33]. Among them, only five, i.e., A_1g_ + E_g_ + 3F_2g_, are Raman active. These modes are due to the motion of O ions into the tetrahedral and octahedral sites. Generally, in AB_2_O_4_ ferrites, the modes above 600 cm^−1^ belong to AO_4_ tetrahedra, while those below 600 cm^−1^ to octahedral BO_6_ [34]. The A_1g_ mode is due to the symmetric stretching of oxygen atoms along metal–oxygen bonds in the tetrahedral coordination. E_g_ is due to the symmetric bending of oxygen with respect to the metal ion in AO_4_ unit and F_2g_ (1) is due to translational movement of the tetrahedron. F_2g_ (3) is caused by asymmetric bending of oxygen and F_2g_ (2) is due to asymmetric stretching of metal and oxygen of octahedral group. The inversion causes the appearance of a doubling of the Raman structures in the spectrum, particularly evident for the A_1g_ mode. The intensities of the modes are related with the inversion degree [35]. Magnesium ferrites are inverted spinel-type systems and there is accordance to consider the main Raman feature in the range between 600–730 cm^−1^ as composed by two bands associated to stretching of the tetrahedral cages, one at lower energies (around 660 cm^−1^) due to Fe–O bonds and the second at higher energies (around 710 cm^−1^) due to Mg–O bonds [36].

The Raman spectra for pure MW and SG MgFe_2_O_4_ are reported in Figure 4. The Raman sampling has been performed in different sample regions, always producing the same results. From both sample types, spectra presented the expected five Raman modes for MgFe_2_O_4_.

For the MW sample, the whole Raman yield was reduced with respect to the SG counterpart and the intensity ratio of the bands was different. For the SG sample, the second order Raman band was observed at around 1350 cm^−1^ (see arrows in Figure 4). Conversely, this feature was not observed for MW, possibly because of the presence of the Raman fingerprint for disordered carbon structures, with two broadened bands at about 1320 and 1590 cm^−1^, i.e., the well-known D and G bands of carbon systems (see arrows in Figure 4) [37]. This result might be ascribed to a residual of citric acid combustion during synthesis.

In addition, for the MW sample a background signal increasing at lower energies was present, probably due to fluorescence from some traces of impurity.

In agreement with the literature and our previous works on ferrites systems [27,33,34,35,36], the Raman structures in the range 600–800 cm^−1^ were analyzed in order to obtain the inversion degree, i.e., the amount of Fe cations in the tetrahedra replacing Mg. The data have been processed by the best fitting procedure using a sum of Lorenztian curves (Appendix A). The derived values for the inversion degree were in good agreement with those obtained from XRPD measurements and Rietveld refinements, indicating that the SG sample was characterized by a 30% higher amount of Mg substituting for Fe inside the octahedral sites with respect to MW.

The doping with silver introduced relevant changes in Raman spectra for both MW and SG samples. For all the doped samples, the Raman spectra were systematically characterized by broadened and weaker features with respect to the pure ones. In addition, the doping influenced samples homogeneity (Figure 5 and Figure 6). 

This was particularly true for the SG-1 sample, as also evidenced by optical microscopy observations (not shown) in which a colored texture, with light blue parts, was observed. The Raman spectra taken from these regions (see arrows in the SG-1 panel in Figure 5) showed Raman fingerprint of hematite overlapped to the expected Raman spectrum for magnesium ferrite, even if this was weakened and broadened. Outside these regions, the spinel-type Raman spectrum could be observed (see spectrum b in the SG-1 panel in Figure 5). Two representative spectra obtained from the SG-3 sample are reported in Figure 5: one (spectrum c) was clearly caused by a high disorder degree in the magnesium ferrite. The second one (spectrum d) was characterized by an enhanced Raman activity between 300–600 cm^−1^ with broadened features peaked at around 630 and 390 cm^−1^ overwhelming the main A_1g_ Raman mode for spinel-type ferrites. This behavior was observed also for other samples, and could be associated to a sort of SERS effect due to Ag nanoparticles plasmonic effects.

The results from SG-1T and SG-3T treated samples evidenced slight changes with respect to the untreated ones, even if a Raman total yield increase could be observed associated with an increase of the Raman band at around 700 cm^−1^. This can be due to an increase of crystallite sizes for SG samples and thus to a more ordered structure caused by the thermal treatment (see Table 2).

Similar results have been obtained from MW samples (Figure 6). The Raman spectra from doped samples denoted an increase of the Raman activity between 400–600 cm^−1^, in particular for MW-3T. A fluorescence signal centered in the infrared region, far away from the Raman energy region already described, can be evidenced in MW samples, even if with different intensities. MW-1T appeared inhomogeneous: in fact, three different behavior, associated to the three different spectra reported (hematite, broadened signal and ferrite, from top to the bottom, Figure 6) were evidenced. These results are in agreement with the XRPD data, that thermally treated samples evidenced the hematite phase.

### 3.4. Antibacterial Activity

The MgFe_2_O_4_-based nanomaterials were tested for their antibacterial activity towards *E. coli* ATCC25922 and *S. aureus* ATCC25923, representative strains of Gram-negative and Gram-positive bacteria, respectively, two types of strains normally used in antimicrobial susceptibility testing. For both species, antibacterial activity was calculated from the area of inhibition of bacterial growth (Figure 7A) caused by the diffusion of the nanoparticles through the gel mesh and visible as dark halos surrounding the nanomaterial deposition sites (Figure 7B). Except for MW-1, all nanoparticles inhibited the bacterial proliferation to different extents, of both microbial species. SG samples appeared generally more active than MW-based materials, and the antibacterial activity was generally more pronounced towards *S. aureus* than towards *E. coli*.

MW specimens were non-significantly or mildly effective without thermal treatment; however, a remarkable improvement was observed upon heating. In *E. coli*, MW particles effectiveness was MW-1T > MW-3T > MW-3, while in *S. aureus* effectiveness was MW-3T > MW-1T > MW-3, thus confirming the importance of the thermal treatment.

SG particles effectiveness against *E. coli* seems not dependent on the thermal treatment, as the inhibitory activity was classified as SG-1 > SG-3 > SG-3T > SG-1T. In *S. aureus*, SG effectiveness was SG-3T > SG-3 > SG-1 > SG-1T, suggesting that the Ag amount is probably the most effective determinant.

A clear relationship between the physico-chemical characteristics and the antibacterial properties of the ferrite samples is not easily identifiable. However, some considerations can be made. The higher antibacterial activity of SG samples could be linked to different factors. First, the nanometric sizes of silver particles (about 10 nm, Table 2), unchanged also after the thermal treatment, in the ideal range for bacteria targeting. In addition, by comparing the growth inhibition cause by undoped and doped materials, the presence of silver into the spinel lattice as a substituting ion has a pronounced positive effect on the antibacterial activity, which increases with the amount of dopant. These results are in line with previous observations [22]. The silver particles were also well distributed into the SG samples (Appendix A) without aggregation, which is detrimental for antibacterial activity. This is not the case of MW samples, in which the crystallite sizes of silver were almost double for all samples, forming noticeable aggregates (Appendix A) which is not evident in the corresponding SG samples. The activity of MW particles was mild, particularly with lower Ag doping, indicating that, in this type of material, the amount of Ag plays a role. The effect dramatically increased, however, with thermal treatment, that not only caused a slight increase of the crystallite sizes of spinel and silver particles, but also induced the nucleation of hematite, itself displaying well-known antibacterial properties [38], which could have a synergistic effect with silver. Thus, we can speculate that the most critical parameter is not the silver amount but, instead, is the size of the silver nanoparticles [22] and their distribution (into the spinel lattice or as metal aggregates), which should be properly tuned to control the antibacterial activity.

## 4. Conclusions

Antimicrobial resistance represents an important global health issue and the research for alternative antibacterials is a pressing theme. In this contest, the silver-based magnesium spinel ferrites synthesized here were demonstrated to have interesting antibacterial properties, particularly against two of the most critical bacteria, *E. coli*. and *S. aureus*. The proper choice of the synthesis methodology, sol–gel or microwave-assisted, allowing the formation of nanometric metallic silver that was well distributed into nanometric spinel particles, was the key factor for obtaining a satisfactory antibacterial effect. The presence of silver as the dopant ion into the spinel lattice could help to increase the antibacterial activity, provided that the grains of powders were nanometric. When, instead, silver nucleated as a separated phase, its homogeneous distribution onto the powder is mandatory, avoiding detrimental agglomerates. The additional thermal treatment could be useful for allowing the nucleation of hematite, itself possessing antimicrobial activity, providing a synergistic effect with silver.

We think that our work could provide an advancement in research on alternative antibacterial materials.

## Figures and Tables

**Figure 1 materials-14-02859-f001:**
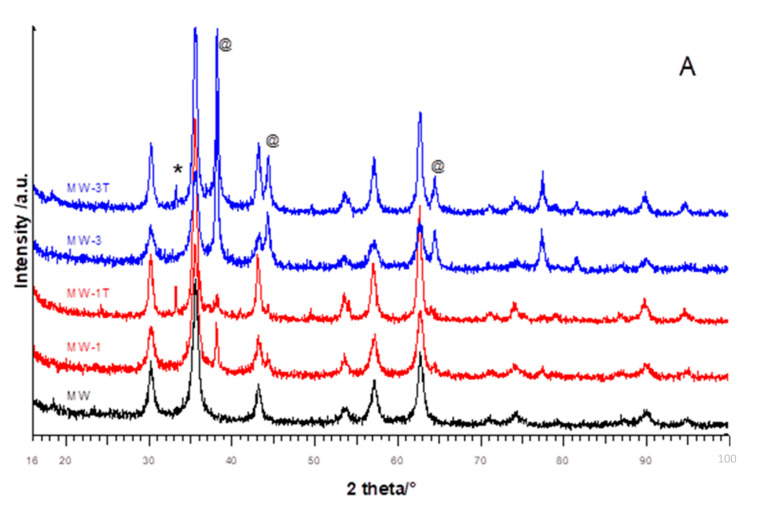
XRPD patterns of pure, doped and thermally treated samples from (**A**) MW and (**B**) SG syntheses. Symbols mark the main peaks of hematite Fe_2_O_3_ (*) and Ag (@); all the remaining peaks pertain to the main spinel phase.

**Figure 2 materials-14-02859-f002:**
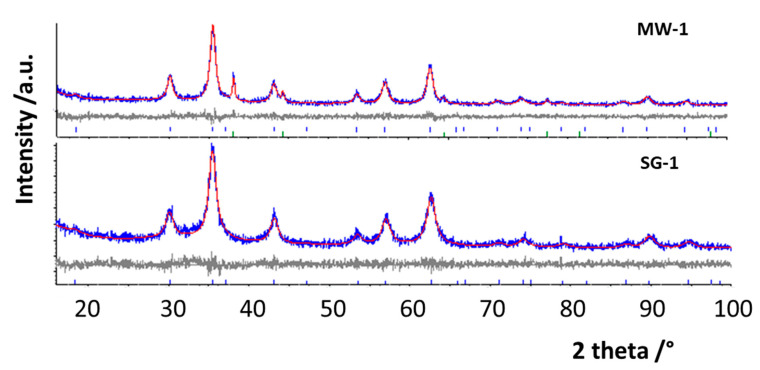
Rietveld refinements of the patterns of the samples indicated above the graphs, chosen as examples. The experimental curve (blue) is compared with the calculated one (red). On the bottom, the difference curve (grey) and the bars at the positions expected for the reflections for the different phases are also shown.

**Figure 3 materials-14-02859-f003:**
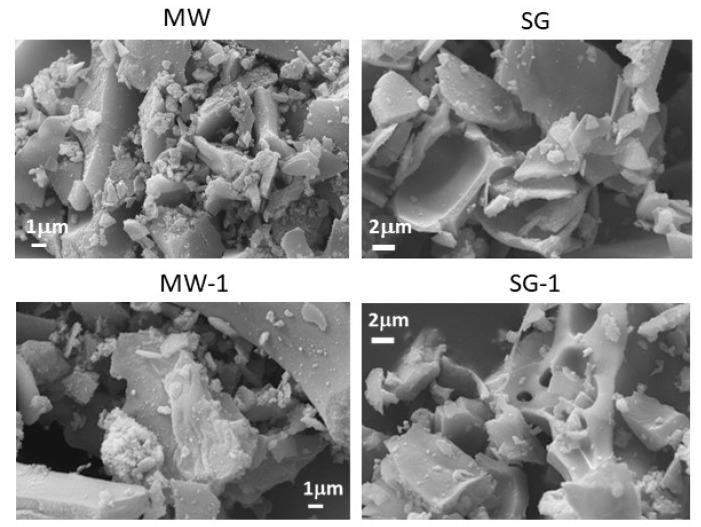
SEM images with samples identity given above each image. The magnification is 10 kx for all images.

**Figure 4 materials-14-02859-f004:**
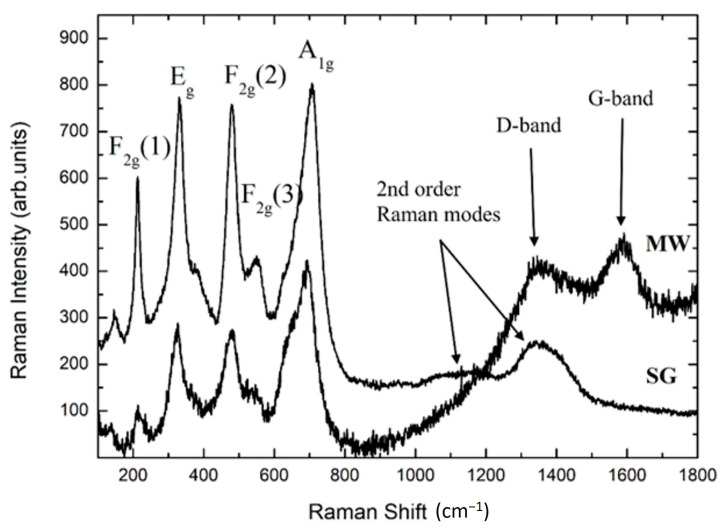
Raman spectra of pure MgFe_2_O_4_ samples.

**Figure 5 materials-14-02859-f005:**
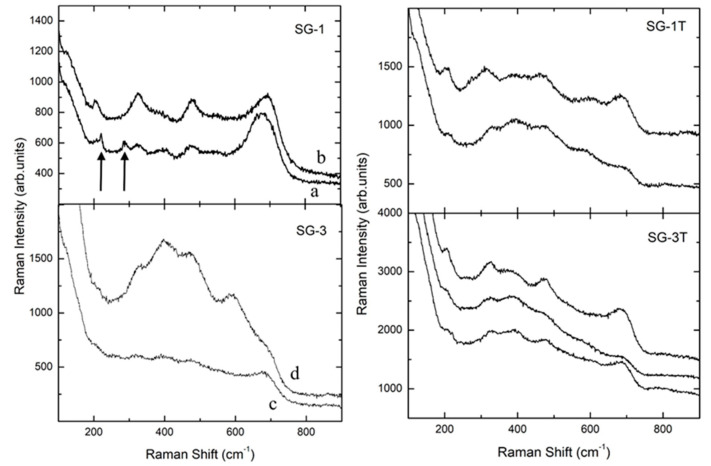
Raman spectra of doped and heat-treated sol–gel samples. For each sample, representative spectra from different sample regions are reported.

**Figure 6 materials-14-02859-f006:**
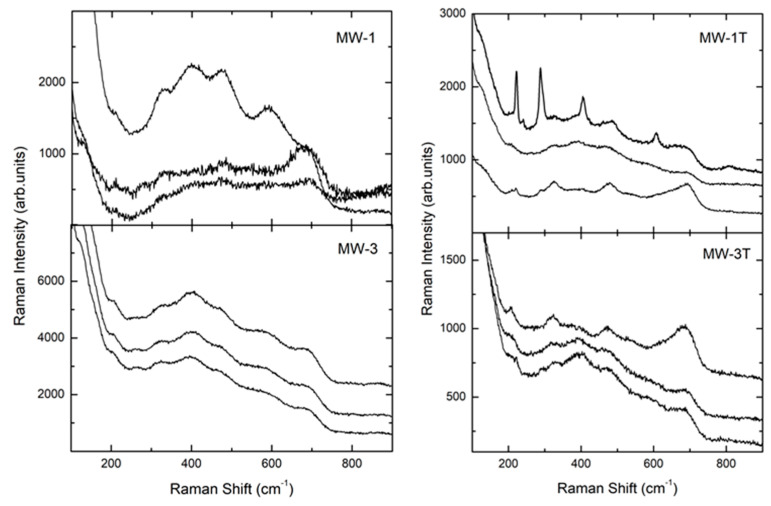
Raman spectra of doped and heat-treated microwave samples. For each sample, representative spectra from different samples regions are reported.

**Figure 7 materials-14-02859-f007:**
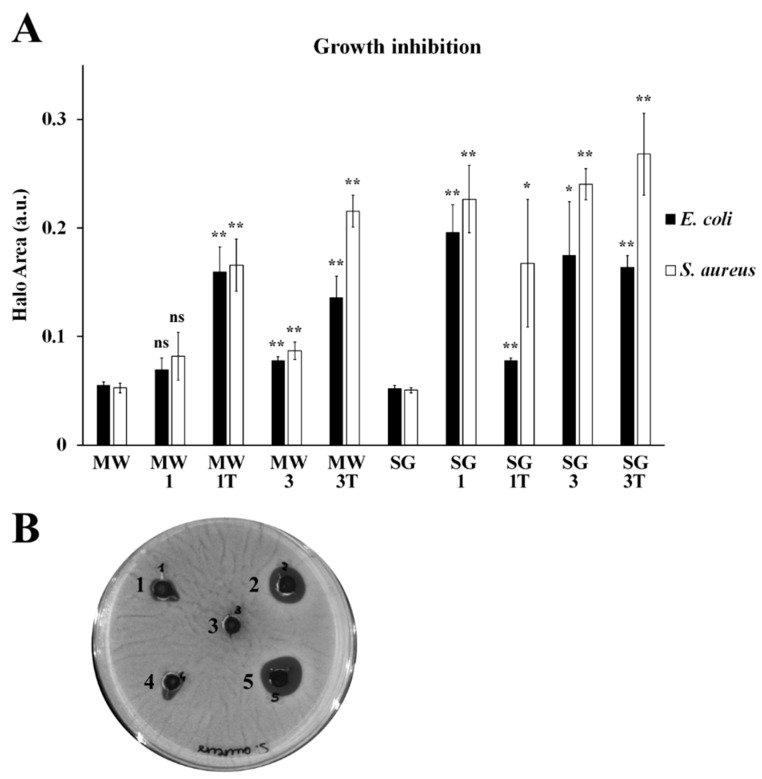
(**A**) Antibacterial activity of MgFe_2_O_4_-based materials against *E. coli* (black columns) and *S. aureus* (white columns), evaluated by growth inhibition assays. The graph displays the averages from three biological replicates, with error bars representing the standard deviation. The identity of each sample is provided on the bottom. On the *y*-axis, the area of growth inhibition is expressed in arbitrary units (a.u.). Undoped microwave and sol–gel materials, indicated with MW and SG, respectively, were used as control for statistical analysis for each bacterial strain. Significant growth inhibition (** *p* value < 0.01; * *p* value < 0.05; n.s: non-significant; two-tailed Student’s *t* test) was observed for all material except for microwave particles doped with Ag_0.1_. (**B**) Representative image of *S. Aureus* growth inhibition assay by MW particles. (1) MW-3; (2) MW-1T; (3) MW; (4) MW-1; and (5) MW-3T.

**Table 1 materials-14-02859-t001:** Main structural parameters from Rietveld refinement for the MW samples.

Samples	a/Å	Cry Size Spinel (nm)	Cry Size Silver (nm)	Cation Distribution on T and O Sites	I	wt.% Fe_2_O_3_	R_wp_/ GoF
**MW**	8.3734(13)	13(1)	-	[Mg_0.64_Fe_0.36_]_T_ [Fe_1.64_Mg_0.36_]_O_	0.36	-	14.0/ 1.15
**MW-1**	8.3765(12)	12(1)	22(2)	[Mg_0.29_Fe_0.71_]_T_ [Fe_1.28_Mg_0.71_]_O_	0.71	-	12.8/ 1.06
**MW-3**	8.3777(12)	11(1)	24(1)	[Mg_0.25_Fe_0.75_]_T_ [Fe_1.25_Mg_0.75_]_O_	0.75	-	12.1/ 1.07
**MW-1T**	8.3868(7)	19(1)	35(1)	[Mg_0.22_Fe_0.78_]_T_ [Fe_1.22_Mg_0.78_]_O_	0.78	5	14.0/ 1.08
**MW-3T**	8.3861(7)	19(1)	36(2)	[Mg_0.23_Fe_0.77_]_T_ [Fe_1.23_Mg_0.77_]_O_	0.77	3	12.5/ 1.06

**Table 2 materials-14-02859-t002:** Main structural parameters from Rietveld refinement for the SG samples.

Samples	a/Å	Cry Size (nm)	Cry Size Silver (nm)	Cation Distribution on T and O Sites	I	wt.% Fe_2_O_3_	R_wp_/ GoF
**SG**	8.3792(7)	18(1)	-	[Mg_0.53_Fe_0.47_]_T_ [Fe_1.53_Mg_0.47_]_O_	0.47	-	16.4/ 1.23
**SG-1**	8.3849(20)	8(1)	-	[Mg_0.23_Fe_0.77_]_T_ [Fe_1.13_Mg_0.77_Ag_0.1_]_O_	0.77	-	13.4/ 1.03
**SG-3**	8.3930(25)	8(1)	11(1)	[Mg_0.26_Fe_0.74_]_T_ [Fe_1.26_Mg_0.74_]_O_	0.74	-	13.3/ 1.07
**SG-1T**	8.3885(3)	48(1)	9(1)	[Mg_0.26_Fe_0.74_]_T_ [Fe_1.26_Mg_0.74_]_O_	0.74	4	14/ 1.05
**SG-3T**	8.3888(4)	44(1)	11(1)	[Mg_0.25_Fe_0.75_]_T_ [Fe_1.25_Mg_0.75_]_O_	0.75	-	10.5/ 1.08

## Data Availability

The data are available upon request.

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
