# Peer review of "Silver Doped Magnesium Ferrite Nanoparticles: Physico-Chemical Characterization and Antibacterial Activity"

_materials, 2021, doi:10.3390/ma14112859_

Round 1
Reviewer 1 Report
This manuscript from Fantozzi et al. shows their investigation of the physical and chemical properties and antibacterial activities from the silver doped MgFe2O4 nanoparticles. They also show the eventual thermal treatment could be useful for providing a synergistic effect in ani-bacterial activity. Antimicrobial resistance is an important topic for alternative antibacterials. In this paper, the authors studied the synthesized silver-based magnesium spinel ferrites and demonstrated the useful antibacterial properties. They also showed the prepared nanoparticles have antibacterial properties against two of the most critical bacteria (E. coli. and S. aureus), and the effect of the silver dopant for enhancement of the antibacterial activities.
I would recommend this work for publication, but I also suggest the authors consider addressing some minor concerns, listed below.
1) In figure 1b, can the authors commented on the absence of the Fe2O3 peak of the SG-3T compared to Figure 1a?
2) Line 79 and the rest of the manuscript, please use “microwave-assisted” instead of “microwave assisted”
3) In figure 2, the x-axis labeled as “2-TH Degrees”, please consider adding the full name here to avoid confusion.
In future 2, the label on the top figure, MW-1, is too small to see. Please update the figure.
Also, the x-label of the bottom figure is missing
4) Figure 3, the MW samples and the SG samples are shown, but the images of the two types of materials are different, 10kx and 20kx, I suggest authors make the magnification the same for a fair comparison
5) Please consider labeling the second-order Raman bands on the figure for clear comparison for both MW and SG curves in Figure 4
6) In figure 5 and 6, two or more curves are in the same subplots, I suggest authors use arrows to label each curve to avoid confusion
7) Both the figure 7a and 7b are low image quality. Please update those images for better readability
Less...
Reviewer 2 Report
The paper entitled “Ag-doped MgFe2O4 nanoparticles: physicochemical characterization and antibacterial activity” presents the fabrication of Ag-doped Magnesium Ferrite nanoparticles and studied their physicochemical characterization as a novel approach to antibacterial studies. Although several papers on silver as an antibacterial agent were reported, this paper seems to be interesting, requiring some revisions. It could be suitable after addressing the notified comments below.
Remove abbreviations or formulas or short forms from the title for the reader’s understanding.
- This manuscript is not easy to understand because of inappropriate sentence formation. For example, in the abstract, sentences 10 and 11, "In particular, MgFe2O4, a partially inverted spinel phase, could find application in medicine thanks to a marked antibacterial activity attributed to the generation of reactive oxygen species." I think the author should rewrite such phrases to be readable in the abstract.
- In the introduction, I suggest citing some articles relevant to silver against antibacterial efficacy. Journal of Photochemistry & Photobiology, B: Biology 169 (2017) 124–133, Mol. Pharmaceutics 2015, 12, 2289−2304.
- The author should mention how much percentage of silver content in the samples SG, SG-1, and SG-3, for good antibacterial activity. Upon increasing Ag percentage, any changes in the antibacterial activity?
- I suggest the authors providing the TEM images
- The antibacterial activity in this study should be compared with some previous studies materials to showcase the advantages of the material.
- In section 3.4, antibacterial studies for the effectiveness of MW particles are based on thermal treatment. I'm wondering all MW particles are synthesized under the same condition. Why showed different antibacterial efficacies? The authors should provide any other reason for changing antibacterial activity.
- In section 3.4, the authors mentioned amount of Ag in SG particle synthesis gives good antibacterial activity and not dependent on the thermal treatment. I was wondering that why the same formula will not be applicable in MW particles.
- Several small paragraphs, it’s better to merge them into meaningful paragraphs in the manuscript.
- Please add instrument info to your devices in the manuscript. (e.g., SEM (Type, company, location?)).
- To enrichment of the manuscript quality, the authors are advised to include a separate section in results and discussion to compare the efficiency of this study with the previous work.
- There are some errors throughout the paper (e.g., Attention should also be given to the use of E. coli and S. aureus, representative strains of Gram-negative and Gram-positive bacteria. The author-written wrong in the manuscript.
- It is required to provide a comprehensive evaluation of antibacterial efficacy. I suggest adding the ROS results with effect to Ag doping in the manuscript. No clear detail in the legend of figure 7B, about the bacterial strain.
Reviewer 3 Report
The reviewed manuscript presents the physical, chemical, and biological characterization of the Ag-doped MgFe2O4 nanoparticles. The main topic of the research has been to develop the novel material with enhanced, concerning a spinel alone, antibacterial activity for medical purposes.
The topic can be considered novel and unique. The problem of the antibacterial shield in the area of implants has been considered very often, The first attempts were antibiotics incorporated into any material, then nanometals or some phosphates, including mainly silver or its salts, and many others. This attempt is fascinating and unique as far as I know. It is original as I know no similar papers of these or other authors, and I often search for some antibacterial solutions for coatings which is the main specialization of my group.
The contribution of the research described in the paper is obvious. The use of nanometals, antibiotics, and phosphates has several disadvantages and the present research focuses on silver bound by other compounds, as chelates or spinels. This paper certainly may make a new pulse to looking for stable Ag-containing materials, safer and more resistant to, e.g. elevated temperature.
The paper is well organized, well written, very clear to a reader, prepared in almost perfect English.
The conclusions are supported by the obtained results and justified by a deep scientific discussion. In particular, it has been shown that the synthesized silver-based magnesium spinel ferrites have demonstrated significant antibacterial properties, against E. coli. and S. aureus. The synthesis methodology, sol-gel or microwave, allowing the formation of nanometric metallic silver well distributed onto nanometric spinel particles.
Reviewer 4 Report
This paper introduces the synthesis of Ag doped MgFe2O4 NPs by MW and SG, and studied the anribacterial effect for the E-coli and S. aureus.
However, this manuscript insufficient to the Nanomaterials due to the unelaborate preparation of manuscript.
There is not existing for the exact purpose of the research.
1). Why synthesized by a great different method of MW and SG?
2). If synthesized, detailed discussion must be necessary for the difference and must suggest a suitable mechanism inducing a different result.
3). There is no explanation at all how prepare (or calculate) the values of Table 1 and Table 2.
4). Thereis no result for the SEM/EDS at all.
5). There is no discussion on the result of for the effectivness of antibacterial effect SG-3T>SG-3>SG-1>SG-1T .
If the Ag amount is probably the most effective determinant, why synthesized complicated spinel structure by MW and SG?
etc.
Therefore, I can't recommend this manuscript to Nanomaterials in the present state because it has no impact.
Round 2
Reviewer 2 Report
Accept
Reviewer 4 Report
Although I feel miss to an insincere response without separate answers to the Reviewer comments, I accept this manuscript for the publication to the Nanomaterials because they revised sufficiently.